# Biodegradation of Aflatoxin B_1_ in the Baijiu Brewing Process by *Bacillus cereus*

**DOI:** 10.3390/toxins15010065

**Published:** 2023-01-12

**Authors:** Guoli Xue, Yanjun Qu, Dan Wu, Shuyuan Huang, Yuqing Che, Jing Yu, Ping Song

**Affiliations:** School of Food Science and Pharmaceutical Engineering, Nanjing Normal University, Wenyuan Road, Nanjing 210023, China

**Keywords:** aflatoxin B_1_, biodegradation, *Bacillus*, baijiu, aroma compounds

## Abstract

Aflatoxin is a potent mycotoxin and a common source of grain contamination that leads to great economic losses and health problems. Although distilled baijiu cannot be contaminated by aflatoxin, its presence in the brewing process affects the physiological activities of micro-organisms and reduces product quality. *Bacillus cereus* XSWW9 capable of degrading aflatoxin B_1_ (AFB_1_) was isolated from daqu using coumarin as the sole carbon source. XSWW9 degraded 86.7% of 1 mg/L AFB_1_ after incubation at 37 °C for 72 h and tolerated up to 1 mg/L AFB_1_ with no inhibitory effects. Enzymes in the cell-free supernatant of XSSW9 played a significant role in AFB_1_ degradation. The AFB_1_-degradation activity was sensitive to protease K and SDS treatment, which indicated that extracellular proteins were responsible for the degradation of AFB_1_. In order to investigate the AFB_1_-degradation ability of XSSW9 during the baijiu brewing process, AFB_1_ and XSWW9 were added to grain fermentation (FG-T) and normal grain fermentation without AFB_1_, while normal grain fermentation without AFB_1_ and XSWW9 was used as a control (FG-C). At the end of the fermentation, 99% AFB_1_ was degraded in the residue of fermented grains. The differences of microbial communities in the fermented grains showed that there were no significant differences between FG-T and FG-C in the relative abundance of dominant genera. The analysis of volatile compounds of their distillation showed that the contents of skeleton flavor components was similar between FG-T and FG-C. These results offer a basis for the development of effective strategies to reduce the effect of AFB_1_ on the brewing process and ensure that the production of baijiu is stable.

## 1. Introduction

Mycotoxins are secondary metabolites produced by some fungi that grow on cereals and cereal products. Most mycotoxins have carcinogenic, teratogenic, and mutagenic effects, among which aflatoxin is the most damaging [1]. In addition to their effects on human health, mycotoxins affect the microbial community structure and the metabolic functions of micro-organisms. Aflatoxin is classified as a class I carcinogen by the Cancer Research Institute of the World Health Organization (WHO). It is a highly toxic substance which can damage the liver of humans or livestock and potentially lead to liver cancer and even death in severe cases [2]. More than 20 types of aflatoxins and derivatives have been isolated to date, among which AFB_1_ is the most toxic and carcinogenic [3]. Under favorable growth conditions, mycotoxigenic fungi can lead to contamination during grain production, storage, processing, and distribution [4]. Sorghum is often damaged by rain in the field and severely infected with grain mold, including aspergillus infection and aflatoxin production. Mycotoxin contamination and grain mold of sorghum are considered the most important threats to grain quality and production across the world [4]. In addition, corn is affected by its storage environment (such as temperature and humidity), which is often conducive to the production of AFB_1._

Baijiu is one of the six most popular distilled spirits in the world with a history that dates back thousands of years. Usually, baijiu is produced by solid-state fermentation of sorghum or multiple grains with daqu as the fermentation starter, after which the fermented grains are distilled to produce baijiu [5]. Contamination by *Aspergillus flavus* is usually not a concern in the production of baijiu because mycotoxins have a high boiling point and remain in the solids [6]. In laboratory simulations of the sake brewing process, AFB_1,_ aflatoxin B_2_(AFB_2_), aflatoxin G_1_(AFG_1_), and aflatoxin G_2_(AFG_2_) were added to raw sorghum, but none of these four compounds were detected in the fresh spirits after fermentation and distillation [7]. Similarly, when fermented barley sochu mash was artificially contaminated with 13 mycotoxins, no tested mycotoxins were detected in the distillate obtained using small instruments [8]. Finally, risk monitoring at critical control points for baijiu production in well-controlled plants showed that aflatoxin and Ochratoxin A (OTA) levels were lower than the maximum limit of the Chinese national standard (the standard being that aflatoxin content in grain should not exceed 5 μg/kg [9]).

Although there are no mycotoxins in distilled liquor, the fermentation process of baijiu is always conducted in a warm and humid environment, and there is a potential risk of mycotoxin contamination in the production process of baijiu. Accordingly, fungal toxins in the fermentation process may affect the growth of distilling micro-organisms and thus affect the quality of liquor. For example, zearalenone, deoxynivalenol, and AFB_1_ have negative effects on the growth and viability of *Saccharomyces cerevisiae* [10,11,12]. Similarly, both zearalenone and T-2 inhibited the growth of *S. cerevisiae*, and these toxins may have implications for the fermentation industry [13,14,15]. In addition, the fungal pathogens *Fusarium* and *Alternaria* on wheat affect the growth of *Pseudomonas* after the production of toxins [16,17,18]. Overall, the production of mycotoxins and other secondary metabolites by fungi plays an important role in the development and competition of organisms within the biofilm matrix [19,20]. Therefore, degrading aflatoxin during the fermentation of baijiu can further improve the stability of the microbial community during the fermentation process and ensure baijiu’s quality.

In past decades, interest in the biological control of mycotoxin pollution using selected micro-organisms has greatly increased, which provides an attractive alternative for eliminating toxins to safeguard the quality of food and feed [21]. The use of microbes as well as isolated enzymes for the removal of aflatoxins from food and feed is an efficient, specific, and environmentally friendly decontamination strategy [22]. Strains of several species, including *Lactobacillus acidophilus* [23,24], *Pichia caribbica* [25,26], *Streptomyces rimosus* [27], and *Aspergillus niger* [28] have been used. In many cases, the main way for lactic acid bacteria to reduce AFB_1_ is cell adsorption, not degradation [29]. Furthermore, the absorption seems to be reversible, which leads to the incomplete removal of AFB_1_ [30]. Conversely, fungal species face challenges in their potential application for the detoxification of AFB_1_ due to their long degradation time, inadaptability to typical food fermentation conditions, production of pigments, and other problems. Many bacterial species of the genus *Bacillus* produce antibacterial agents, and most can be easily cultured in low-cost media. In addition, its spore-forming characteristics help it survive under adverse conditions and maintain a high population density in challenging environments [31]. For these reasons, *Bacillus* is becoming an attractive candidate for the biological control of pathogenic fungi and the prevention of mycotoxin production. Previous studies have shown that *Bacillus* spp. can degrade ochratoxin [32], zearalenone [33], and deoxynivalenol [34]. In addition, some reports show that *Bacillus* can inhibit the growth of *A. flavus* and reduce aflatoxin contamination [35]. Similarly, *B. subtilis* JSW-1 can inhibit the growth of *A. flavus* with an inhibition rate of 58.3% after 72 h and a degradation rate of AFB_1_ that reaches 67.2% [36]. Similarly, *B. pumilus* E-1-1-1 can degrade 89.55% in 12 h [37], while *B. licheniformis* CFR1 can degrade 94.7% in 24 h [38].

Previously, there were few reports on the use of micro-organisms to degrade mycotoxins in baijiu brewing materials or control the mycotoxins produced by daqu micro-organisms. In this study, we explored the effects of using AFB_1_-degrading strains in the baijiu brewing process. We isolated bacteria capable of degrading AFB_1_ from daqu and evaluated the ability of the isolates to degrade AFB_1_. The microbial community dynamics and metabolic profile in the baijiu brew were analyzed. Overall, this study represents the first step in understanding the metabolism and function of AFB_1_-degrading strains in brewing liquor, and it may help improve the quality of baijiu.

## 2. Results and Discussion

### 2.1. Isolation of Microbial Strains for AFB_1_ Degradation

A strain with a stable and remarkable performance of AFB_1_ degradation was screened by subculturing it six times in a primary liquid medium with coumarin as the sole carbon source. This strain, named XSWW9, could degrade more than 85% of 1 mg/L AFB_1_ after 2 d of cultivation in a fermentation medium under static conditions at 37 °C.

We cloned and analyzed the complete 16S rRNA gene of strain XSWW9. Phylogenetic analysis indicated that XSSW9 was a strain of *Bacillus cereus*. The 16S rRNA gene of XSSW9 was 99.24% similar to that of *Bacillus cereus* strain ATCC 14579 (Figure 1a). The colonies appeared pale yellow to yellow on potato dextrose agar and other nutritive media (Figure 1b). XSWW9 was able to hydrolyze aesculin, but not starch. The optimum pH was 6.0–8.0, and the strain could grow at 20–50 °C with an optimum growth temperature range of 30–37 °C.

Next, XSSW9 was cultured with 1 mg/L AFB_1_, and the biomass of the strain and AFB_1_ degradation were investigated. After 36 h of rapid growth, the biomass of XSSW9 reached a peak and stabilized, and after 48 h, the biomass gradually dropped. It was suggested that the metabolism of the strain was extremely active in the 24–48 h window. The period of AFB_1_ degradation coincided with this time interval. The concentration of AFB_1_ dropped quickly as microbial growth increased at the beginning of the lag phase, and more than 85% of AFB_1_ was degraded within 72 h (Figure 1c).

### 2.2. Adaptability of B. cereus XSSW9 to the Temperatures Found in Baijiu Fermentation Pits

During the fermentation of baijiu, the grains are fermented in fermentation pits at temperatures ranging from 20 to 50 °C. Since the survival status of micro-organisms is greatly affected by temperature, we investigated whether *B. cereus* XSSW9 can survive normally in this environment and degrade toxins. A batch test of AFB_1_ degradation by XSWW9 fermentation broth with an initial AFB_1_ concentration of 1 mg/L was carried out at 20–50 °C (Figure 2). The optimal temperature for the growth of XSWW9 was 37 °C. Accordingly, XSWW9 presented the highest AFB_1_ degradation ratio at 37 °C as the removal of AFB_1_ by either adsorption or biodegradation is dependent on the biomass. In addition, the biomass and degradation rate of AFB_1_ were lower when the strain was grown at 50 °C. It is possible that the high temperature inhibited the growth of the strain; thus, the degradation rate of AFB_1_ was also significantly lower. The biomass of the strain and the degradation rate of AFB_1_ increased alongside increasing temperatures between 20 and 37 °C. In this range, increased temperatures may have promoted the bioavailability of organic compounds, which thereby facilitated the biodegradation process [39]. Overall, this strain has a high growth and AFB_1_ degradation rate at temperatures found in baijiu fermentation pits.

### 2.3. Degradation of AFB_1_ by the Cell-Free Supernatant, Cell Supernatant, and Cell Extracts of B. cereus XSWW9

AFB_1_ degradation due to cell-free supernatant, cell supernatant, and cell extracts of XSSW9 cultured in fermentation medium without AFB_1_ for 48 h was investigated by conducting an additional incubation of these fractions with 1 mg/L AFB_1_ in the dark at 37 °C for 72 h (Figure 3). AFB_1_ was shown to be stable after 72 h of incubation in pbs at 37 °C. A significant reduction (*p* < 0.0001; 87.04% AFB_1_ degradation) was observed when AFB_1_ was treated with the cell-free supernatant of XSWW9. In contrast, the bacterial cells and cell extracts showed relatively low AFB_1_ degradation ratios (20.4% and 41.4%, respectively). These results suggested that a secreted factor played a major role in AFB_1_ degradation.

Next, protease K treatment was performed to investigate whether the degradation of AFB_1_ is enzymatic. The AFB_1_ degradation ratio of the cell-free supernatant decreased to 59.8% after treatment with protease K, and it decreased to 12.9% after treatment with protease K combined with SDS. Similarly, the AFB_1_ degradation ratio of the cell extracts decreased to 27.3% after treatment with protease K, and it decreased to 11.8% after treatment with protease K combined with SDS. These results indicated that enzymes or other proteins might play important roles in the degradation of AFB_1_ by the strain XSWW9 [40]. Similar findings were reported by Sangare et al. [41] and Yu et al. [42].

The mechanisms of the microbial detoxification of mycotoxins have been classified previously based on binding or degradation [43]. Currently, the most common mechanism applied to remove mycotoxins on an industrial scale is adsorption. However, adsorbed toxins can potentially be released during prolonged incubation periods [44,45]. In strain XSWW9, the proteins secreted by the cells played a major role in AFB_1_ degradation. Therefore, XSWW9 may have greater application potential than available strains.

### 2.4. The Effect of Temperature and pH on AFB_1_ Degradation by the Cell-Free Supernatant of B. cereus XSSW9

The degradation rate of AFB_1_ increased as the incubation temperature increased from 20 °C to 37 °C. However, when the incubation temperature reached 50 °C, the degradation rate of AFB_1_ showed a significant decrease (Figure 4a). These results are consistent with the hypothesis that the factor responsible for mycotoxin degradation in the supernatant of strain XSWW9 is a protein.

Figure 4b shows that AFB_1_ degradation by the cell-free supernatant of XSWW9 changed with the pH, with a maximum at pH 6.0–8.0.

The time profile of AFB_1_ degradation due to the cell-free supernatant was also recorded. Most of the degradation occurred within 48 h, and the degradation rate decreased as the AFB_1_ concentration fell (Figure 4c).

### 2.5. Degradation of AFB_1_ by the Cell-Free Supernatant of XSSW9 during Grain Fermentation

To verify AFB_1_’s removal ability in actual production, external AFB_1_ was added to the zaopei without measurable mycotoxin content to a final concentration of 20 μg/kg, and the normal baijiu production fermenter was set as a blank control. In order to assess the representative degradation rate of AFB_1_ in the grains, three different height grains were selected from the top, middle, and lower layers. Table 1 shows the content of residual AFB_1_. The AFB_1_ degradation rate was close to 100% after 28 days of fermentation in the fermenter, which was significantly higher than in the cell-free supernatant, which was 87.04%. We consider two reasons for this finding. First, the actual AFB_1_ content during fermentation (20 μg/kg) was much lower than that of the cell-free supernatant (1 mg/kg). Moreover, it is likely that the nutrients in the actual production were richer, so cell activity was maintained longer and reached a higher total biomass. At the same time, the upper limit of AFB_1_ degradation in actual production and the controllable degree of actual system contamination needs to be further explored.

### 2.6. Dynamics of Microbial AFB_1_ Degradation during Grain Fermentation

Dynamics within the grain microbial consortium during AFB_1_ degradation were determined using 16S rRNA sequencing. AFB1 and XSWW9 were added to the grain fermentation (FG-T), while normal grain fermentation without AFB1 and XSWW9 was used as control (FG-C). Fermented grains with 20 μg/kg AFB_1_ were tested at four timepoints—0 d (initial fermented grains), 7 d (rapid warming period of wine lees), 14 d (lees stabilization stage), and 28 d (pre-wine making). After quality filtering and assignment, a total of 74,933, 40,129, 31,902, and 35,589 OTUs were generated at each timepoint. Similarly, the data for the control group included 81,557, 36,832, 39,406, and 31,799 OTUs. Figure 5 shows the microbial consortium composition and the eleven most abundant bacterial phyla in the different fermented grains based on the OTU assignments.

The initial control grain samples (0 d) presented the major genera *Weissella* (59.48%), *Acinetobacter* (12.15%), *Enterobacter* (145%), and *Lactococcus* (8.21%). Other classes comprising more than 0.10% included *Kosakonia* (0.34%), *Cronobacter* (4.15%), *Leuconostoc* (0.26%), *Lactobacillus* (2.90%), and *Enterococcus* (2.88%) (Figure 5a). At the same time, the initial grain samples (0 d) contained the major genera *Weissella* (58.81%), *Acinetobacter* (4.41%), *Enterobacter* (14.99%), and *Lactococcus* (7.34%). Five other classes comprised more than 0.10%, including *Kosakonia* (0.41%), *Cronobacter* (4.10%), *Leuconostoc* (0.89%), *Lactobacillus* (1.86%), and *Enterococcus* (2.86%) (Figure 5b). There was no significant difference between the two major strains and their percentages. The principal component analysis (PCA) of microbial fractions in the fermentation process showed that FG-C and FG-T could be divided into four main components, and there were significant differences in the microbial fractions of the same fermentations at different times, while the differences in the microbial fractions of FG-C and FG-T at the same period were not significant (Figure 5c).

In both the control and experimental groups, *Weissella* was predominant during the first seven days of fermentation, but after 14 days of fermentation, the content of *Weissella* decreased significantly, while the content of *Acinetobacter* increased rapidly and gradually became the dominant strain in solid-state fermentation. It is worth noting that the richness and diversity of prokaryotes in the zaopei did not change significantly when fortified bacterial liquid was added. This observation is also consistent with the observation that the aroma compounds of the baijiu produced by the two fermentations were not significantly different (Figure 6). These results indicated that the traditional fermentation process is largely unaffected when the added strains are derived from the daqu.

In the fermentation process of baijiu, the primary micro-organisms are bacteria and actinomycetes, and bacteria play a major role in aroma production. Most of the *Bacillus* species have high proteinase activity and are functional bacteria in the Maillard reaction in baijiu. *Weissella* and *Acinetobacter* are functional bacteria that metabolize ethanol and catalyze ester production during fermentation. In addition, most strains of *Weissella* can grow on methanol as the only carbon source, which is helpful to reduce the methanol content in baijiu.

### 2.7. Analysis of Volatile Aroma Compounds during AFB_1_ Degradation in Grain Fermentation

The aroma compounds in liquor distillated from different fermentations were identified to assess the influence of zaopei bioaugmentation on the volatile profiles of fresh liquor. In the experimental group, 164 volatile compounds were detected, including 58 esters, 23 alcohols, 8 acids, 20 aldehydes, 11 ketones, 3 furans, 5 phenols, and 37 other compounds. Similarly, a total of 169 volatile substances were detected in the blank control, including 60 esters, 23 alcohols, 8 acids, 21 aldehydes, 11 ketones, 3 furans, 6 phenols, and 37 other compounds (Figure 6a,b). The alcohol meter measured 50.7% vol for the experimental group and 51.4% vol for the control group. This also corresponds to the content of ethanol in the GC-MS profiles of both groups.

The flavor characteristics of baijiu are mainly determined by esters, alcohols, aldehydes, ketones, and other volatile compounds. Therefore, we analyzed major volatile compounds in the two groups of baijiu, including ethyl acetate, butyrate, butyl acetate, isobutanol, isoamyl acetate, n-butanol, isoamyl alcohol, ethyl n-hexanoate, ethyl lactate, ethyl caprylate, 1-octen-3-ol, isoamyl lactate, ethyl caprate, phenylacetaldehyde, diethyl succinate, phenethyl acetate, phenethyl alcohol, and ethyl 9-oxononanoate (Figure 6c). Overall, no significant differences in flavor compounds were observed in the two groups of baijiu. As shown in Figure 6, in the experimental group, the largest family of volatile flavors was esters, followed by acids, alcohols, and a small number of ketones, aromatics, and aldehydes. Likewise, the control liquor presented high contents of these six flavor groups. These results indicated that the difference in the content of the main volatile compounds between the two types of baijiu was very small and almost negligible. We can therefore conclude that the addition of the cell-free supernatant of the experimental strain during the baijiu solid-state fermentation process resulted in the degradation of AFB_1_ in the grains without affecting the flavor and taste of the baijiu. This conclusion is also in line with our expectations.

**Figure 6 toxins-15-00065-f006:**
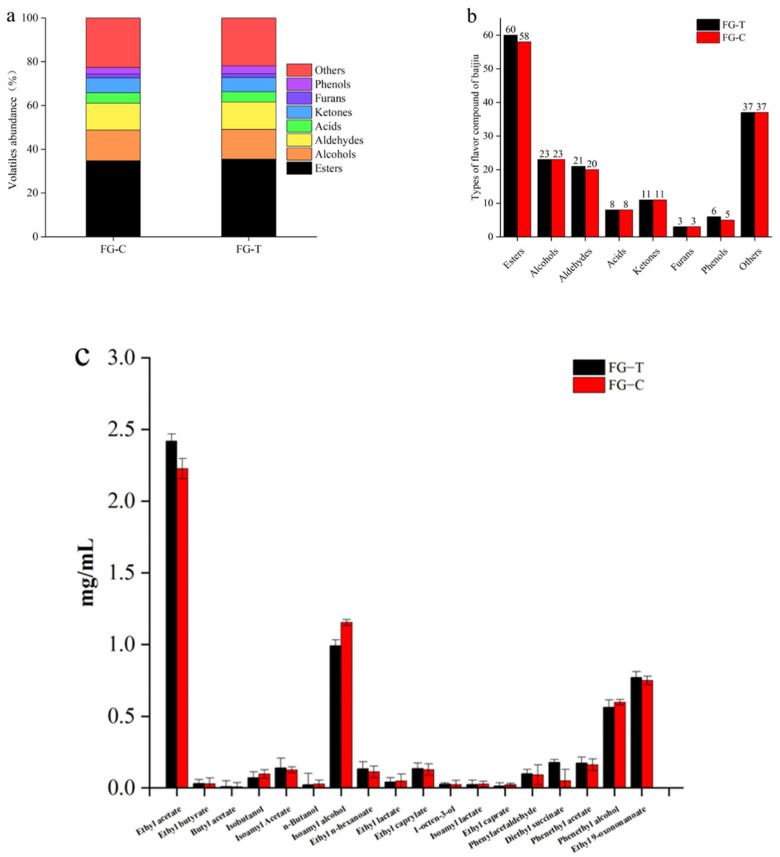
Volatile compounds in baijiu from the control and experimental groups. (**a**) Abundance of volatile compounds in baijiu from the control and experimental groups. (**b**) Content of volatile substances in baijiu from the control and experimental groups. (**c**) Content of major volatile flavor compounds in baijiu from the control and experimental groups.

## 3. Conclusions

We screened a strain of *Bacillus cereus* with a stable degradation ability of AFB_1_ from strong aromatic daqu, and the strain showed very high tolerance to high concentrations of AFB_1_. The strain was blended with normal zaopei for bioaugmentation, and the solid-state fermentation process was analyzed using GC-MS. We found that the strain could efficiently degrade AFB_1_ without affecting the microbial community of zaopei or the aroma profile of baijiu. Thus, the bioaugmentation of zaopei can further improve the safety factor during the fermentation of the grains without affecting the quality of baijiu. In conclusion, we provide a new approach for the degradation of AFB_1_ potentially present in baijiu zaopei. The role of this strain in other foods and feeds needs to be further investigated.

## 4. Materials and Methods

### 4.1. Chemicals and Culture Media

AFB_1_ was purchased from Macklin (Shanghai Macklin Biochemical Technology Co., Shanghai, China).

NB medium was measured in g/L and contained 10 peptones, 3 beef extract powders, and 5 NaCl. The pH was adjusted to 7.4.

Enrichment medium was measured in g/L and contained 0.25 KH_2_PO_4_, 0.003 FeCl_3_·6H_2_O, 0.25 MgSO_4_·7H_2_O, 0.5 KNO_3_, 0.5 (NH_4_)_2_SO_4_, and 0.005 CaCl_2_·2H_2_O. The pH was adjusted to 7.0. Different concentrations of coumarin-ethanol solution were added as needed after sterilization.

Fermentation medium was measured in g/L and contained 10 peptones, 3 beef extract powders, 8.5 NaCl, 1 KH_2_PO_4_, and 1 glucose. The pH was adjusted to 7.0.

### 4.2. Preparation of Daqu Micro-Organism Suspensions

Daqu from a traditional production facility was used as the microbial source. Five grams of daqu material were taken and transferred into 100 mL of sterile water, homogenized with glass beads, shaken for 2 h at 37 °C, and dispersed well to obtain a suspension.

### 4.3. Enrichment and Isolation of Degrading Micro-Organism

Coumarin was used as the sole carbon source to enrich the microbes which can degrade AFB_1_, according to a previously described method [46]. The daqu micro-organism suspensions were seeded into 100 mL of NB medium at 10% inoculum in 250 mL flasks and incubated under static conditions at 37 °C for 3 d. For the first transfer, the culture was transferred into 150 mL of enrichment medium (with a final concentration of coumarin 1.0 g/L) at 10% inoculum in a 250 mL flask at 37 °C and 160 rpm for 3 d. The transfer was conducted repeatedly with gradually increasing concentrations of coumarin for the enrichment cultivation.

After the last enrichment, the culture was serially diluted, 100 μL of diluted solution was spread on an isolation agar medium each time, and then it was cultured at 37 °C for 5 d. The selected colonies were picked repeatedly from the isolation agar plates until apparent purity was reached.

### 4.4. Culture of Micro-Organisms

The selected strain from the isolation agar plates was grown in a fermentation medium at 37 °C for 24 h with shaking at 160 rpm to obtain the fermentation broth.

### 4.5. Degradation Test of AFB_1_ by B. cereus XSSW9

A sample comprising of 800 μL of fermentation (including alternatively cell-free supernatant, washed cells, and cell extract) was combined with 200 μL of AFB_1_ methanol solution (5 mg/L) in a 5 mL EP tube, followed by incubation at 37 °C and 160 rpm for 72 h in the dark. Finally, the residual AFB_1_ concentration was determined. 

### 4.6. Detection of AFB_1_ Concentration

AFB_1_ was extracted from the degradation test sample with an equal volume chloroform three times and evaporated under nitrogen gas. The samples were dissolved in a methanol–water solution (1:1, *v*/*v*) and filtered. Then, the concentration of AFB_1_ was analyzed using HPLC-FLD on a Shimadzu lc-40 Series instrument (Shimadzu, Nanjing, China) equipped with a fluorescence detector. The chromatographic column was a Venusil MP C18 (5.0 μm, 250 mm × 4.6 mm, 5 μm), and it was kept at 40 °C. The mobile phase was methanol:H_2_O (1:1, *v*/*v*) with a flow rate of 1.0 mL/min, and the injection volume was 10 μL. The excitation and detection wavelength were set to 360 and 440 nm, respectively. The degradation ratio was calculated according to the following formula:

degradation ratio = (1 − AFB_1_ peak area following treatment/AFB_1_ peak area in initial sample) × 100%.



### 4.7. Identification of the AFB_1_ Degrading Strain

The selected potential strain was identified using 16S rDNA sequence analysis. The 16S rDNA was amplified by PCR using the universal primers 27F (5′-AGAGTTTGATCCTGGCTCAG-3′) and 1492R (5′-TACGGCTACCTTGTTACGACTT-3′). The sequencing was conducted by Shanghai Majorbio Bio-pharm Technology Co., Ltd. (Shanghai, China). Then, the sequence was compared with known sequences in the GenBank database using the Basic Local Alignment Search Tool (BLAST) program (http://blast.ncbi.nlm.nih.gov, accessed on 27 November 2021). After multiple sequence alignments were performed using CLUSTAL_X 2.1, a phylogenetic tree was constructed using MEGA 11.0.10. 

### 4.8. Preparation of Cell-Free Supernatant, Cell Supernatant, and Cell Extracts of B. cereus XSSW9

After activation in a sterilized fermentation medium, the strain was transplanted into 100 mL of fresh fermentation medium in a 250 mL flask and incubated under dynamic conditions at 37 °C and 160 rpm for 48 h. After centrifugation at 12,000 g for 10 min, the 20 mL cell-free supernatant and cells were collected.

The cells were washed twice and re-suspended in an equal volume of 50 mM Phosphate buffered saline (PBS) (PH6.4) to prepare the cell supernatant. 

The cells were disintegrated using an ultrasonic cell disintegrator (Nanjing Xian’ou Instrument Manufacturing Co., Nanjing, China). After centrifugation at 12,000 g for 10 min at 4 °C, the supernatant was filtered using 0.22 μm syringe filters (Tianjin Keyilon Experimental Equipment Co., Tianjin, China) to produce the cell extracts.

The effect of the protease treatment was determined by exposing the cell-free supernatant and cell extract to 1 g/L protease K (Shanghai Aladdin Biochemical Technology Co., Ltd, Shanghai, China). for 1 h at 37 °C or 1 g/L protease K plus 1% SDS for 6 h at 37 °C [47]. 

### 4.9. The Effect of Temperature and pH on AFB_1_ Degradation by the Cell-Free Supernatant of B. cereus XSSW9

The cell-free supernatant was incubated with 1 mg/L AFB_1_ in the dark at 37 °C for 72 h, and the AFB_1_ concentration was determined at 0 h, 4 h, 8 h, 12 h, 24 h, 48 h, and 72 h. To determine the effect of temperature, the reaction mixture was incubated in the dark at 20, 25, 28, 30, and 37 °C for 72 h.

In the pH tests, initial pH values of the reaction mixture were adjusted to 4.0, 5.0, 6.0, 7.0, 8.0, 9.0, 10.0, and 11.0 with relevant sodium phosphate buffers. The filtrated fermentation medium was adjusted to different pH values and used as a control.

### 4.10. Grain Fermentation (Baijiu Brewing)

A sample comprising 10 kg of sorghum was broken into 4–8 petals, and a weighed sample of 8 kg was suspended in hot water above 85 °C, which corresponded to a moisture content of 58–62%. After 24 h of wetting, the grain was treated with retort steam for 0–60 min, after which 30 vol % of 80–90 °C hot water was added, spread, and left to cool to 25 °C. Then, 10% of daqu, 200 μg of AFB_1_, and 20 mL of cell-free supernatant of XSWW9 were added, mixed well, placed into a fermentation barrel, sealed with film, and fermented for 28 days. The grain fermentation with the added AFB_1_ and XSWW9 was used to test the AFB_1_-degrading ability of XSSW9 during the liquor brewing process, and the samples were named FG-T. The normal grain fermentation without AFB_1_ and XSWW9 was used as a control named FG-C. After 28 days, the grains were distilled in three fractions, and the middle fraction of high-quality baijiu was used for analysis.

### 4.11. Determination of Volatile Compounds

Headspace solid-phase microextraction gas chromatography–mass spectrometry (HS-SPME-GC-MS) was used to determine the volatile compounds [40]. Samples comprising of 200 μL of the liquor were placed into a 10 mL headspace vial with 3 mL of saturated NaCl, 10 μL of internal standard (2-octanol, 0.0822 g/L, Sigma-Aldrich, St. Louis, MO, USA), and a magnetic rotor. They were then placed in a constant temperature water bath with a magnetic stirrer for mixing at 60 °C for 50 min. At the same time, a 50/30 μm DVB/CAR/PDMS fiber (Supelco, Bellefonte, PA, USA) was used to extract the volatile compounds for 45 min at 60 °C. Then, the vial was placed in a thermostatic block stirrer to equilibrate for 10 min at 50 °C and subsequently extracted for 45 min with stirring at 500 rpm. After extraction, the SPME fiber was withdrawn into the needle and immediately inserted into the injection port of the Thermo Fisher Scientific GC–MS(GC/MS-QP2020., Shimadzu Corporation, Kyoto, Japan) system, which was coupled with a trace TR-WAX fused silica capillary column (30 m × 0.25 mm i.d; 0.25 µm film thickness; the inlet temperature was 250 °C; there was no splitting; the column temperature profile was 40 °C and was maintained for 5 min; then, it was ramped up to 230 °C at 5 °C/min and maintained for 5 min; a high purity helium was the carrier gas at a flow rate of 1 mL/min; the connection port temperature was 250 °C; the ionization mode was EI; the electron energy was 70 eV; the ion source temperature was 230 °C; and the scan range was 40–450 amu). The mass spectra of the sample analytes were compared with the standard spectra. The data were processed using Xcalibur™ software (Thermo Fisher Scientific, Waltham, MA, USA). The component content was calculated using the peak area normalization method, and the final compound mass concentration was expressed as the mean ± standard deviation.

### 4.12. Statistical Analysis

All assays were conducted in triplicate unless otherwise specified. Data are presented as the means ± the standard deviations (SD). A Student’s *t*-test was used to perform the statistical analysis. Statistical significance was accepted at a *p* value < 0.05.

## Figures and Tables

**Figure 1 toxins-15-00065-f001:**
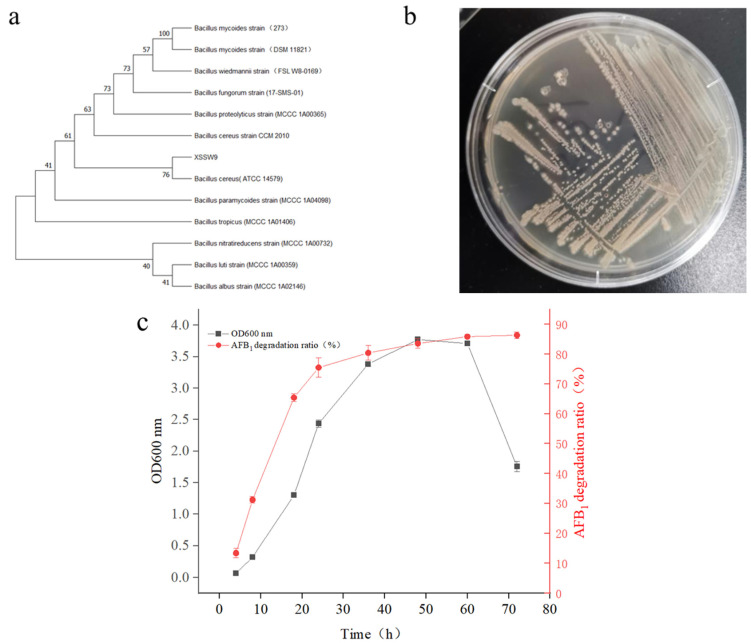
Morphological and phylogenetic characteristics of XSWW9. (**a**) A phylogenetic tree based on 16S rRNA gene sequences created using the neighbor-joining method and showing the relationships among XSWW9 and members of the genus *Bacillus*. Bootstrap values (expressed as percentages of 1000 replications) higher than 50% are shown at branching points. The bar indicates a phylogenetic distance of 0.02 substitutions per nucleotide position. (**b**) Bacterial colonies on PDA at 24 h. (**c**) AFB_1_ degradation ratio (%) and OD600 nm during AFB_1_ degradation by XSWW9. This strain was cultured in a fermentation medium with 1 mg/L AFB_1_ at 37 °C. The data represent the means and standard deviations from triplicate independent experiments.

**Figure 2 toxins-15-00065-f002:**
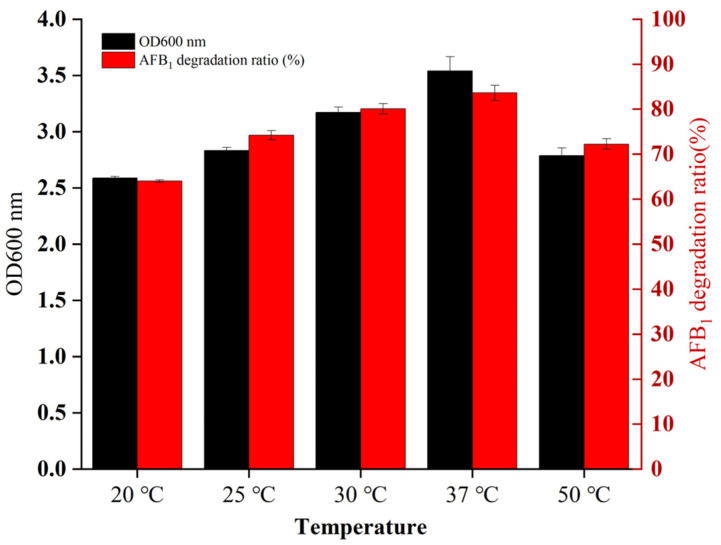
The effect of temperature on the AFB_1_ degradation capacity and biomass growth of XSWW9. The strain XSSW9 was cultured in a fermentation medium with 1 mg/L AFB_1_ at the indicated temperatures. The data represent the means and standard deviations from triplicate independent experiments.

**Figure 3 toxins-15-00065-f003:**
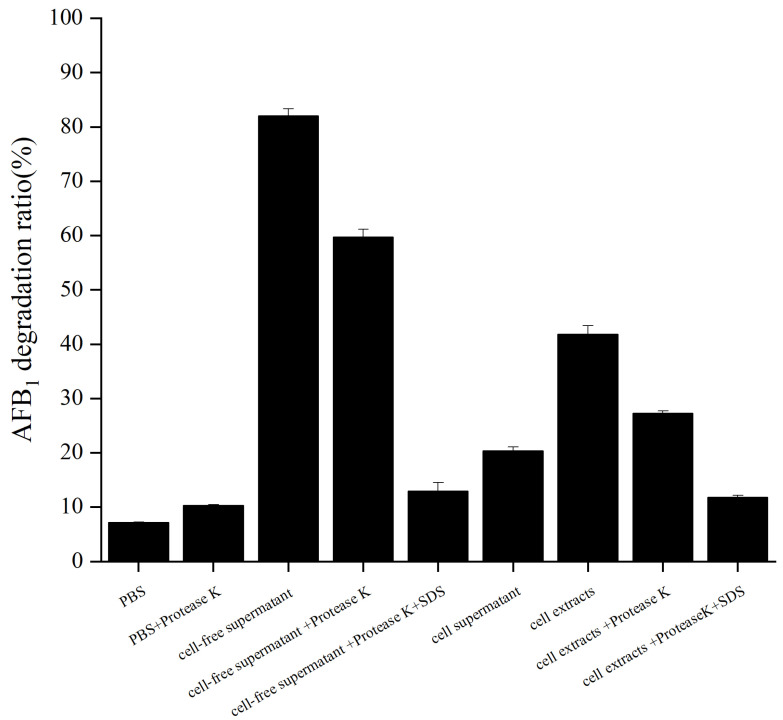
AFB_1_ degradation due to cell-free supernatant, cell supernatant, and cell extracts of XSWW9. The strain XSWW9 was cultured in a fermentation medium without AFB_1_ at 37 °C for 48 h. The cell-free supernatant, cell supernatant, and cell extracts were collected and cultured at 37 °C for 48 h to test their ability to degrade AFB_1_ and the effect of protease K treatment. The data represent the means and standard deviations for triplicate independent experiments.

**Figure 4 toxins-15-00065-f004:**
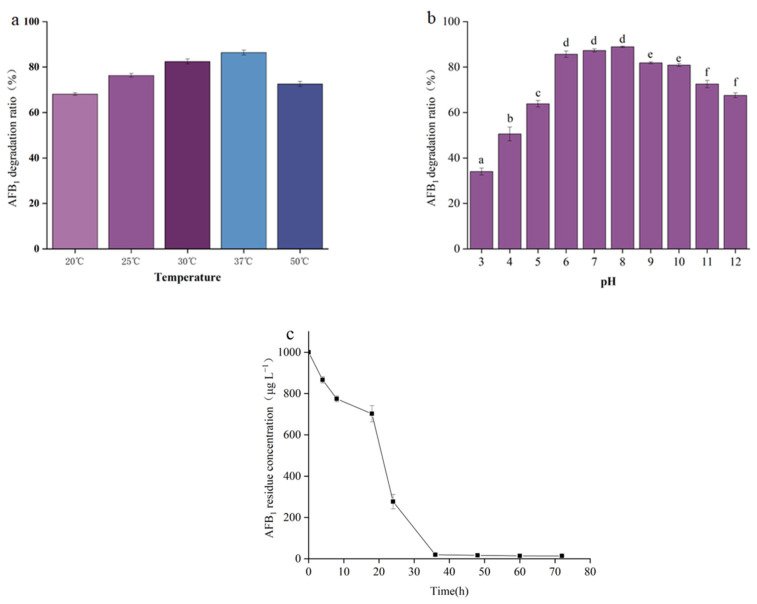
The effects of temperature, pH, and incubation time on AFB_1_ degradation due to the cell-free supernatant of XSSW9. (**a**) AFB_1_ degradation at different temperatures. The cell-free supernatant of XSSW9 was incubated with 1 mg/L AFB_1_ at 20, 25, 30, 37, and 50 °C for 72 h. (**b**) AFB_1_ degradation at different pH. The cell-free supernatant of XSSW9 was incubated with 1 mg/L AFB_1_ at 37 °C with pH 3, 4, 5, 6, 7, 8, 9, 10, 11, and 12 for 72 h. (**c**) Time profile of AFB_1_ degradation due to the cell-free supernatant of XSSW9. The cell-free supernatant of XSWW9 was incubated with 1 mg/L AFB_1_ at 37 °C with pH 8 for 72 h. The data represent the means and standard deviations of triplicate independent experiments. Values with different letters are significantly different.

**Figure 5 toxins-15-00065-f005:**
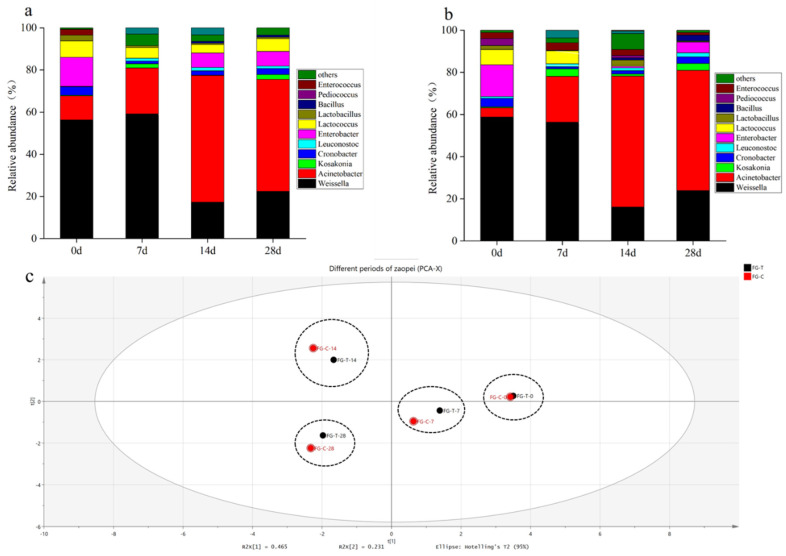
Diversity dynamics of the bacterial community during the solid state fermentation of baijiu. The diversity of bacteria during solid state fermentation of grains was tested at four timepoints—0 d (initial fermented grains), 7 d (rapid warming period of wine lees), 14 d (lees stabilization stage), and 28 d (pre-wine making). (**a**) FG-C. (**b**) FG-T. (**c**) PCA plots of FG-C and FG-T at different fermentation periods.

**Table 1 toxins-15-00065-t001:** The degradation of AFB_1_ in solid-state fermentation.

Samples	AFB_1_ Content (μg/kg)	AFB_1_ Degradation Ratio (%)
Top layer of grains	<0.1	>99%
Middle layer of grains	<0.1	>99%
Lower layer of grains	<0.1	>99%

Note: It can be concluded that AFB_1_ has been completely degraded in the sample when the content of AFB_1_ in the sample is less than 0.1 μg/kg.

## Data Availability

The data presented in this study are available in this article.

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
