# Peer review of "Biodegradation of Aflatoxin B1 in the Baijiu Brewing Process by Bacillus cereus"

_toxins, 2023, doi:10.3390/toxins15010065_

Round 1
Reviewer 1 Report
The manuscript is very well written and organized. The applied methods were adequate for the study objective.
The following comments should be clarified:
Introduction - (AFB1, OT, AFG1, etc.) The complete name of mycotoxins should appear first time in the text with the abbreviation between brackets, after that it can be used only the abbreviation.
Line 228 - replace "optimum" by optimum growth temperature range of...
Results - Why authors selected the concentration of AFB1 0f 1 mg/L? Other concentrations of mycotoxins could be analysed, since concentration is a critical parameter for degradation.
References are not described according to the journal standards.
Reviewer 2 Report
Aflatoxins, especially AFB1, are a serious problem in agricultural and food production, and biological approach for AFB1 degradation is actively developed as a promising and environmentally safe decontamination strategy. Numerous studies report the microbial detoxification of products contaminated with A. flavus and other toxigenic fungi. Nevertheless, studies aimed at searching for new effective aflatoxin biodestructors are of current interest, especially from the point of view of their further practical use on an industrial scale. At the same time, use of microorganisms possessing toxin-degrading activity can have disadvantages, such as production of undesirable metabolites as well as suppression of beneficial microbiota that worsens the quality of resulting products. However, usage of microbial AFB1-catabolizing metabolites, including detoxifying enzymes, allows the avoidance of the drawback of decontamination by living microorganisms.
This manuscript reports a lot of work on isolation, identification and characterization of a Bacillus cereus strain, which effectively removed AFB1 from model mixtures and during fermentation due to secreted proteinaceous metabolites. Authors convincingly demonstrated that this strain and the supernatant of its cell-free culture did not negatively influence on the composition and ratio of distilling bacteria involved in the fermentation process as well as did not affect volatile compounds of baijiu. Thus, decontamination of sorghum grain based on XSWW9 may be applied to prevent possible losses in the baijiu quality.
I have no essential critical comments concerning manuscript conception, used methods, results presentation and conclusions.
Some minor comments, suggestions, and corrections, which could be easily made by authors, are listed below.
Lines 9-10. Why are different units used to denote the same concentration? The same cases are in the text (e.g., lines 221, 229, 247).
Line 15. The description of the control treatment causes a confusion. Please correct to ", ...while normal grain fermentation without AFB1 and XSWW9 was used as control (FG-C) ", as it is in the text (line 189).
Lines 38-41. I would suggest the authors to add several references related to statements presented here.
Lines 53-54. Does OTA mean ochratoxin A? Please provide full name.
Lines 74-75. Not strains but species are indicated. It would be better to write: "Strains of several species, including ……. have been used."
Line 137. Change "… simple…" to "sample"
Line 162. Please indicate the pH value of the buffer.
Line 233. Please consider "… coincided with this time interval…" as an alternative version.
Line 244. I would suggest "… at temperatures ranging from 20 to 50 C." instead of "…a temperature of approximately 20-50 C."
Figure 3. Figures should be self-sufficient and completely understandable to a reader. Please provide a full name for PBS (no corresponding abbreviation is in the text) and unify the names of the columns inside the figure and in the figure caption: cell supernatant = washed cells? Again, intracellular cell extract = cell extract?
Line 313. Please indicate pH value, not only temperature for the subfigure C. Was the pH value within the optimal range 6.0-8.0?
Line 320. "…three different grains..."?? Do you meant "… three different grain samples..."? Or may be "grain portions…"?
Lines 339-400. I suggest changing the current version of the phrase fragment to: "…addition of the cell-free supernatant of the experimental strain during the baijiu solid-state fermentation process resulted in degradation of AFB1…"
I recommend accepting the manuscript for publication after minor revisions.
